Retrospective study and implementation of a low-cost LAMP-turbidimetric assay for screening α0-thalassemia (SEA deletion): preventing and controlling Hb Bart’s hydrops fetalis syndrome in Thailand

Jomoui Wittaya wittayaj@g.swu.ac.th 1 2
Saknava Kanokkorn 3
Prechatrammaruch Kanokpron 3
Ondee Yanticha 3
1 Department of Pathology, Maha Chakri Sirindhorn Medical Center, Faculty of Medicine, Srinakharinwirot University , Ongkharak , Nakhon Nayok , Thailand
2 Clinical Research Center, Faculty of Medicine, Srinakharinwirot University , Ongkharak , Nakhon Nayok , Thailand
3 Faculty of Medicine, Srinakharinwirot University , Ongkharak , Nakhon Nayok , Thailand
Sezgin Efe
Electronic publication date: 2024 Feb 27
Publication date: 2024
Volume: 12
Electronic Location ID: e17054
Received 2023 Dec 14; Accepted 2024 Feb 14
Copyright: ©2024 Jomoui et al.
Copyright year: 2024
Copyright holder: Jomoui et al.
License: This is an open access article distributed under the terms of the Creative Commons Attribution License, which permits unrestricted use, distribution, reproduction and adaptation in any medium and for any purpose provided that it is properly attributed. For attribution, the original author(s), title, publication source (PeerJ) and either DOI or URL of the article must be cited.
License URL: https://creativecommons.org/licenses/by/4.0/

Keywords: Biomedical, Loop-mediated isothermal amplification, Turbidimetric, SEA deletion, Thalassemia

Funding: HRH Princess Mahachakri Sirindhorn Medical Center, Faculty of Medicine, Srinakharinwirot University (Contract No. 162/2566) This study was supported by a research grant from HRH Princess Mahachakri Sirindhorn Medical Center, Faculty of Medicine, Srinakharinwirot University (Contract No. 162/2566). The funders had no role in study design, data collection and analysis, decision to publish, or preparation of the manuscript.

==============================
Homozygous α0-thalassemia (SEA deletion) or Hb Bart’s hydrops fetalis syndrome is a significant public health issue in Thailand and Southeast Asia. A prevention and control program has been implemented in this region. This study focuses on retrospective laboratory data collected between January 2021 and April 2023 at a single center. Additionally, we developed a low-cost LAMP-turbidimetric assay to propose in the screening strategy. A total of 3,623 samples underwent screening tests (MCV, MCH, and DCIP), including 1,658 couple screenings (84.25%) and 310 single pregnant screenings (15.75%). Negative screenings, which did not require further investigation, were found in 75.51% for couple screenings and 46.58% for single pregnant screenings. At hemoglobin (Hb) analysis identified 129 couples which had fetuses at risk of severe thalassemia, whereas molecular analysis during the retrospective period revealed 210 samples with different genotypes. These remaining samples were validated using the low-cost LAMP-turbidimetric assay to detect α0-thalassemia (SEA deletion). The developed LAMP turbidimetric assay demonstrated a sensitivity and specificity of 100% (36/36 × 100) and 97.7% (170/174 × 100), respectively, when compared with gap-PCR. Furthermore, we propose a strategy involving the addition of the low-cost LAMP-turbidimetric assay before performing the gold standard. This strategy represents a cost-saving of USD 2,608 based on 210 samples that required DNA analysis. Finally, the developed LAMP turbidimetric assays offer advantages such as reduced time, workload, cost savings, no need for highly developed instruments, and a straightforward interpreting process. Therefore, implementation of LAMP assays into routine settings would be improve the efficiency of prevention and control program for severe thalassemia disease in this region.

Introduction

Thalassemia is a genetic disorder caused by a decrease or absence of globin chains related to the α-globin gene (chromosome 16) and/or the β-globin gene (chromosome 11) (Fucharoen et al., 1998; Weatherall & Clegg, 2001). The severity of clinical manifestations varies among genotypes and phenotypes. In Thailand, approximately six hundred thousand people are affected by this genetic disease. Thalassemia carriers make up about 30–40% of the Thai population (Fucharoen et al., 1998; Panich, Pornpatkul & Sriroongrueng, 1992). In Thailand, a prevention and control program for severe thalassemia has been implemented, addressing conditions such as homozygous α0-thalassemia, homozygous β0-thalassemia, compound heterozygous β0-thalassemia, and Hemoglobin (Hb) E (Fucharoen et al., 1998). Homozygous α0-thalassemia, or Hb Bart’s hydrops fetalis syndrome, is a severe form caused by the absence of the α-globin gene, leading to the non-production (α0-thalassemia) of α-globin chains in the fetus. Moreover, pregnant women with this condition face an increased risk of severe maternal complications. In Thailand and Southeast Asia, α0-thalassemia (SEA deletion) is the most prevalent, whereas Thai deletion and Filipino deletion have very low incidence rates  (Fucharoen et al., 1998; Chaibunruang et al., 2013).

The laboratory investigation strategy was initiated with a combined screening protocol involving the osmotic fragility (OF) test, red blood cell indices (mean corpuscular volume (MCV) or mean corpuscular hemoglobin (MCH)), and dichlorophenolindophenol (DCIP). Individuals with positive screening results underwent further investigation with Hb analysis. Thalassemia genotypes were confirmed through molecular DNA analysis. Although this screening protocol is simple, cost-effective, straightforward, and highly sensitive, it still presents a high rate of false positives, leading to the need for confirmation with molecular diagnosis (Jomoui et al., 2017a; Fucharoen et al., 2004; Sanchaisuriya et al., 2005). The identification of α0-thalassemia carriers is crucial for the prevention and control program of these diseases in the region. Currently, the gold-standard diagnostic test is the gap-PCR technique (polymerase chain reaction), which identifies DNA deletions in thalassemia (Chaibunruang et al., 2013). However, this method has limitations, including high costs, time consumption, and dependence on specialists and specific equipment. As a result, it cannot be performed in general hospitals in Thailand, especially in community hospitals where resources are limited (Chaibunruang et al., 2013). Several studies have reported the addition of screening tests before molecular diagnosis, such as the immunochromatographic (IC) strip test and Loop-mediated isothermal amplification (LAMP) with various detection methods (Prayalaw, Fucharoen & Fucharoen, 2014; Winichagoon et al., 2015; Jomoui et al., 2022; Chomean et al., 2018; Wang et al., 2020). However, the IC strip test has a higher false positive rate when compared to LAMP assays, which are molecular-based. Therefore, molecular LAMP screening tests could be evaluated and implemented in the prevention and control program for severe α0-thalassemia.

Currently, LAMP assays are employed for genetic screening due to their ability to amplify the LAMP target up to 109 times in approximately one hour, utilizing isothermal incubation. This results in high sensitivity and specificity, as demonstrated in several studies (Jomoui et al., 2022; Chomean et al., 2018; Wang et al., 2020). Moreover, interpretation can be conducted using various techniques, such as naked eye colorimetric assessment, turbidity measurement, lateral flow dipstick analysis, gel electrophoresis, and fluorescent detection (Zhang, Lowe & Gooding, 2014). One of the most cost-effective and rapid methods is the naked eye turbidimetric assay, which observes the LAMP product through the formation of an insoluble cloudy white sediment (magnesium pyrophosphate). This sediment is formed by the binding of pyrophosphate ions and magnesium ions. This technique is particularly intriguing and offers several advantages compared to other reported LAMP assays (Garg et al., 2021; Yang et al., 2012). The advantages of turbidity assays are their cost-effectiveness with cheapest, naked-eye interpretation, and no need post-LAMP steps such as lateral flow dipstick and gel electrophoresis. Furthermore, colorimetric pH indicators change may be influenced by different DNA components sample, while turbidity is not affected (Zhang, Lowe & Gooding, 2014). Consequently, this study aims to develop a LAMP-turbidimetric assay and proposes its implementation for the prevention and control of severe α0-thalassemia (SEA deletion) in Thailand.

Materials and Methods

Subjects and specimens

Ethical approval for this study was obtained from the Institutional Review Board of Srinakharinwirot University, Thailand (SWUEC/E/M-098/2565E). This study, we use leftover specimen that informed with broad consent form with the first collecting specimen. The retrospective study data were collected from an ongoing thalassemia screening program for pregnant women and their husbands between January 2021 and April 2023 at MahaChakri Sirindhorn Medical Center, Faculty of Medicine, Srinakharinwirot University, Thailand. All patient information was limited to laboratory data and fully anonymized with research codes. The criteria for sorting data were dependent on laboratory investigation levels, including screening tests, Hb analysis, and DNA analysis during the same period. A total of 3,623 samples were recruited at the screening testing level, including 1,658 couple screenings (84.25%) and 310 single pregnant screenings (15.75%) (Table 1). Additionally, 129 couples (258 samples) with positive screening results, indicating a fetus at risk of severe thalassemia, were recruited for the Hb analysis level (Table 2). Furthermore, DNA analysis was conducted on 210 samples during the same period of the retrospective study. For the sample size calculation, validation should use at least 203 samples, following the formula Zα2P(1-P)/e2, with Zα = 1.96, P = 0.05, and e = 0.03; when P (prevalence) = 0.05 for α0-thalassemia in Thailand (Jomoui et al., 2022; Charan & Biswas, 2013). Therefore, we recruited our remaining 210 DNA samples, which were deemed suitable for the validation. Additionally, a total of 210 leftover DNA specimens were utilized to evaluate the developed LAMP-turbidimetric assay for the detection of α0-thalassemia (SEA deletion).

Table 1 Thalassemia screening and risk assessment of severe thalassemia disease in fetus during January 2021 to April 2023 at single center (n = 3, 623).

N and P indicate negative, and positive.

Couples screening (N = 1,658 couples; 84.25%)	
Parent I	Parent II	N
(3,316)	Possible risk of severe thalassemia disease in fetus	
MCV*	MCH*	DCIP*	MCV*	MCH*	DCIP*			
N (86.4 ± 3.3)	N (29.2 ± 1.3)	N	N (86.6 ± 3.2)	N (29.1 ± 1.2)	N	447 (x2)	No risk of severe thalassemia disease	
N (86.8 ± 3.4)	N (29.3 ± 1.3)	N	P (75.8 ± 6.5)	P (24.5 ± 2.6)	N	393 (x2)	No risk of severe thalassemia disease	
N (86.3 ± 3.2)	N (29.2 ± 1.8)	N	P (72.4 ± 6.7)	P (24.0 ± 2.4)	P	379 (x2)	No risk of severe thalassemia disease	
N (82.4 ± 1.7)	N (27.8 ± 0.9)	P	N (86.3 ± 3.4)	N (29.1 ± 1.3)	N	33 (x2)	No risk of severe thalassemia disease	
N (82.3 ± 1.4)	N (27.7 ± 0.6)	P	P (74.0 ± 2.4)	P (24.6 ± 1.1)	P	13 (x2)	Compound heterozygous β0 - thalassemia and Hb E	
P (75.9 ± 6.1)	P (24.5 ± 2.2)	N	N (82.7 ± 1.9)	N (27.8 ± 0.6)	P	16 (x2)	Compound heterozygous β0-thalassemia and Hb E	
P (76.1 ± 5.6)	P (24.7 ± 2.3)	N	P (76.6 ± 6.1)	P (24.7 ± 2.5)	N	83 (x2)	Homozygous α0-thalassemia
Homozygous β0-thalassemia	
P (75.7 ± 6.4)	P (24.5 ± 2.6)	N	P (72.4 ± 6.1)	P (24.0 ± 2.1)	P	202 (x2)	Homozygous α0-thalassemia
Homozygous β0-thalassemia
Compound heterozygous β0 - thalassemia and Hb E	
P (73.7 ± 6.5)	P (24.6 ± 2.3)	P	P (73.6 ± 5.7)	P (24.5 ± 2.0)	P	92 (x2)	Homozygous α0-thalassemia
Homozygous β0-thalassemia
Compound heterozygous β0-thalassemia and Hb E	
Single pregnant women screening (N = 310 single; 15.75%)	
MCV	MCH	DCIP	Paternal screening required	N (307)	Possible severe thalassemia disease in fetus	
N (86.7 ± 3.4)	N (29.2 ± 1.2)	N	No	143	No risk of severe thalassemia disease	
P (76.4 ± 6.1)	P (24.7 ± 2.5)	N	Yes	84	Have a risk of severe thalassemia disease	
N (82.2)	N (27.7)	P	Yes	1	Have a risk of severe thalassemia disease	
P (73.2 ± 6.1)	P (24.4 ± 2.2)	P	Yes	79	Have a risk of severe thalassemia disease	
Notes.

* MCV, MCH, and DCIP indicate mean corpuscular volume, mean corpuscular hemoglobin, and Dichlorophenol Indophenol Precipitation.

Table 2 Risk of severe thalassemia in fetus based on Hb analysis during January 2021 to April 2023 (n = 129).

Risk of severe thalassemia in fetus	Parent I	Parent II	N (129)	
Homozygous α0-thalassemia
89.15% (115/129)	A2A	A2A	55	
EE	A2A	26	
EA (Hb E<25%)	A2A	10	
A2A	A2A (Hb A2>3.5%)	9	
EE	EA<25%	3	
A2A	A2ABart’sH	3	
CSEA (Hb E<25%)	A2A	3	
A2A	A2FA	2	
EE	EE	1	
EE	CSEA (Hb E<25%)	1	
EE	EFA	1	
EA (Hb E<25%)	CSEA (Hb E<25%)	1	
Homozygous α0-thalassemia and
Homozygous β0-thalassemia
0.78% (1/129)	A2A (Hb A2>3.5%)	A2A (Hb A2>3.5%)	1	
Homozygous α0-thalassemia and
Compound heterozygous β0-thalassemia /Hb E
2.33% (3/129)	EA (Hb E<25%)	A2A (Hb A2>3.5%)	1	
EE	A2A (Hb A2>3.5%)	1	
EA (Hb E<25%)	EE/EF	1	
Compound heterozygous β0-thalassemia/Hb E
7.75% (10/129)	EA (Hb E>25%)	EF	1	
EA (Hb E>25%)	EE/EF	1	
CSEA (Hb E>25%)	EE/EF	1	
EA (Hb E>25%)	A2A (Hb A2>3.5%)	6	
A2A (Hb A2>3.5%)	EA (Hb E>25%)	1	

Routine thalassemia testing

The hematological parameters (MCV/MCH) were obtained using the hematology automation Sysmex XN3000 (Sysmex, Kobe, Japan). The DCIP test for detecting Hb E was performed using a KKU-DCIP kit. Hb analysis was conducted through capillary electrophoresis (Capillarys 2; Sebia, Lisses, France). Identifications of common α-thalassemia (α0-thalassemia: SEA, THAI, FIL, MED, 20.5 kb deletions, and α+-thalassemia: 3.7 and 4.2 kb deletions) and two common non-deletional α-thalassemias (Hb Constant Spring and Hb Paksé) were routinely performed using gap-PCR and allele-specific PCR methods as described elsewhere  (Jomoui et al., 2023; Fucharoen & Fucharoen, 1994; Charoenwijitkul et al., 2019). For one case of rare α0-thalassemia Chiang Rai (–CR) was also investigated with convention multiplex gap-PCR described elsewhere (Jomoui et al., 2023).

Development of LAMP-turbidimetric assay

The LAMP assay for the detection of α0-thalassemia (SEA deletion) was developed using a turbidimetric assay. LAMP primers, including F3, B3, FIP, and BIP, set in this study were described elsewhere  (Jomoui et al., 2022). The loop probe (LP) used in this study was protected according to petty patent submission number 2203001284 (Thailand). For the LAMP reaction mixture, 25 µl was utilized, containing 1.5 µl of genomic DNA (20–50 ng/µl), 2.5 µl of 10X isothermal amplification buffer, 1.5 µl of 100 mM MgSO4, 5 µl of 5 mM dNTPs, 4.0 µl of 5 M Betaine, 0.5 µl of 10 µM each primer of F3, B3, and LP, 1.0 µl of 40 µM each primer of FIP and BIP, 1 µl of Bst 2.0 DNA Polymerase (8,000 units/mL) (New England Biolabs, Ipswich, MA, USA), and the remaining volume was distilled water. The LAMP mixture was incubated at an isothermal temperature of 65 °C for 40 min. The assay contains insoluble cloudy-white sediments (magnesium pyrophosphate) that change the characteristic from clear to turbid, observable by the naked eye. This turbidity is formed by the binding of pyrophosphate ions and magnesium ions. The specificity assays of the turbidimetric LAMP assay to detect α0-thalassemia (SEA deletion) were compared with gel electrophoresis, as shown in Fig. 1.

Figure 1 The specificity of developed LAMP turbidimetric assays for the detection of α0-thalassemia (SEA deletion) with gel electrophoresis related to different genotype.

Screening for α0-thalassemia (SEA deletion) using LAMP-turbidimetric assay

Re-screening for α0-thalassemia (SEA deletion) was evaluated using 210 leftover DNA specimens from a retrospective collection with various thalassemia genotypes in couples at risk of severe thalassemia. This was conducted as a blinded trial with the LAMP-turbidimetric assays, as described above, and the results were compared with those of the conventional α0-thalassemia PCR assay, as summarized in Table 3. Furthermore, the sensitivity, specificity, NPV (negative predictive values), and PPV (positive predictive values) of the turbidimetric LAMP assay were calculated. The strategy for implementing the developed turbidimetric LAMP assay in routine service was proposed for the prevention and control of severe α0-thalassemia in Thailand, as illustrated in Fig. 2. Additionally, the cost-effectiveness of the protocol was calculated for implementation in any population. In this study, we demonstrated the cost difference between the previous strategy and the new strategy for recruiting 210 samples in molecular testing (refer to Fig. 2).

Table 3 The recruited parents at DNA analysis (n = 210) with results of LAMP turbidimetric assays for α 0 -thalassemia (SEA deletion) compared with conventional PCR .

Screening	Hb Types	Hb A2/E	Hb F	α-globin	β-globin	LAMP turbidity	N (210)	
+/-	A2A	2.5 ± 0.3	0.1 ± 0.2	αα/αα	βA/βA	N	23	
αα/αα	βA/βA	FP	2	
-α3.7/αα	βA/βA	N	45	
-α4.2/αα	βA/βA	N	1	
αCSα/αα	βA/βA	N	1	
αPSα/αα	βA/βA	N	2	
-α3.7/-α3.7	βA/βA	N	13	
-α3.7/αCSα	βA/βA	N	1	
–SEA/αα	βA/βA	P	21	
+/-	A2A>3.5	5.5 ± 0.8	1.7 ± 3.1	αα/αα	BT/βA	N	16	
αα/αα	BT/ βA	FP	1	
-α3.7/αα	BT/βA	N	2	
-4.2α/αα	BT/βA	N	2	
+/-	CSA2A	2.2 ± 0.3	0.1 ± 0.1	αCSα/αα	βA/βA	N	5	
-α3.7/αCSα	βA/βA	N	1	
+/-	A2FA	3.5	21.2	-α3.7/αα	βA/βA	N	1	
+/-	A2ABartH	1.1 ± 0.4	0.6 ± 0.1	–SEA/-α3.7	βA/βA	P	4	
–THAI/-α3.7	βA/βA	N	1	
+/+	EA	21.3 ± 4.7	0.5 ± 1.5	αα/αα	βE/βA	N	4	
-α3.7/αα	βE/βA	N	3	
αCSα/αα	βE/βA	N	2	
αPSα/αα	βE/βA	N	2	
-α3.7/-α3.7	βE/βA	N	1	
-α3.7/-4.2α	βE/βA	N	1	
–SEA/αα	βE/βA	P	7	
–CR/αα	βE/βA	FP	1	
–SEA/-α3.7	βE/ βA	P	1	
–SEA/αCSα	βE/βA	P	1	
+/+	CSEA	21.2 ± 2.5	0.7 ± 0.7	αCSα/αα	βE/βA	N	1	
αCSα/αCSα	βE/βA	N	1	
-α3.7/αCSα	βE/βA	N	3	
+/+	EE	95.7 ± 4.4	2.8 ± 3.0	αα/αα	βE/βE	N	21	
-α3.7/αα	βE/βE	N	12	
-α3.7/-α3.7	βE/βE	N	1	
-α3.7/αCSα	βE/βE	N	1	
–SEA/αα	βE/βE	P	1	
–SEA/-α3.7	βE/βE	P	1	
+/+	EFA	47.9 ± 13.2	15.3 ± 12.8	αα/αα	βA/βA	N	2	
-α3.7/-α3.7	βA/βA	N	1	

Figure 2 The cost-effectiveness of the proposed LAMP turbidimetric assays for detecting α0-thalassemia (SEA deletion) compared with LAMP colorimetric assays and conventional protocol.

This was also demonstrated with 210 samples in our study.

Results

In accordance with the prevention and control program for severe thalassemia, we present retrospective data from January 2021 to April 2023. Table 1 displays thalassemia screening, including MCV, MCH, and DCIP. Among the 3,623 samples, 1,658 couples (84.25%) and 310 single pregnant women (15.75%) were screened. Among the 1,658 couples, a total of 1,252 (75.51%) were found to have no risk of the three targeted severe thalassemia diseases in their fetus. This negative screening included four different patterns, where at least one parent exhibited all negative screening results (-, -, -) for MCV, MCH, and DCIP, respectively. The remaining 406 couples (24.49%) had a fetus at risk of severe thalassemia. Of these, 29 couples (1.75%) were at risk of compound heterozygous β0-thalassemia and Hb E, with two screening patterns including (-, -, + / +, +, +) and (+, +, - / -, -, +). Additionally, a total of 83 couples (5.01%) with the pattern (+, +, - / +, +, -) had a fetus at risk of homozygous α0-thalassemia and homozygous β0-thalassemia. The remaining 294 couples (17.73%) had a fetus at risk of three severe thalassemia types, including homozygous α0-thalassemia, homozygous β0-thalassemia, and compound heterozygous β0-thalassemia/Hb E. This group was related to two screening patterns, namely (+, +, - / +, +, +) and (+, +, + / +, +, +). In contrast, single pregnant screening excluded 143 out of 310 (46.13%) pregnancies with no risk of severe thalassemia diseases. However, the remaining 164 out of 310 (52.90%) pregnancies still had a fetus at risk of severe thalassemia diseases, as indicated by positive screening results.

Hb analysis data were recorded for couples with a fetus still at risk of severe thalassemia from January 2021 to April 2023. About 409 couples with a positive risk assessment at screening underwent further investigation with Hb analysis. However, approximately 280 couples (68.46%) were excluded from the positive screening group during this Hb analysis. The remaining 129 couples had a fetus at risk of severe thalassemia. Table 2 displays the Hb analysis and risk assessment of severe thalassemia in the fetus. Of these, 115 couples (89.15%) were predominantly at risk of only homozygous α0-thalassemia, with 12 different Hb type patterns. One couple (0.78%) had a fetus at risk of both homozygous α0-thalassemia and homozygous β0-thalassemia. Three couples (2.33%) had a fetus at risk of homozygous α0-thalassemia and compound heterozygous β0-thalassemia/Hb E, with three different patterns. The remaining 10 couples (7.75%) had five different patterns, indicating a fetus at risk of all three severe thalassemia types. Furthermore, homozygous α0-thalassemia was found to be the most common risk in fetuses at our centers (119 out of 129 couples, 92.25%). These cases were commonly further investigated with DNA analysis in the next step.

DNA analysis was performed to confirm thalassemia genotypes for the final diagnosis. From January 2021 to April 2023, samples were identified for thalassemia genotypes, and the results are illustrated in Table 3. The screening test, Hb types, and DNA analysis were conducted based on routine laboratory procedures. A total of nine groups based on Hb types with positive screening represented several α-thalassemia and β-thalassemia genotypes in each group. In this study, 38 samples with the α0-thalassemia gene were identified, including 36 samples with SEA deletion, and one sample each with THAI deletion and Chiang Rai (CR). The positive rate of the α0-thalassemia gene was 18.10% (38/210 samples) in molecular DNA analysis.

The developed LAMP-turbidimetric assay was evaluated for detecting α0-thalassemia (SEA deletion) with 210 samples recruited, and the results were summarized in Table 3. All 36 samples with α0-thalassemia (SEA deletion) had a positive result in the LAMP-turbidimetric assay. A total of 170 samples without α0-thalassemia (SEA deletion) tested negative for the developed LAMP-turbidimetric assay. However, the remaining four samples without α0-thalassemia (SEA deletion), including two of (αα/αα, βA/βA), one of (αα/αα, βT/βA), and one of (–CR/αα, βE/βA), represented cross reactions and were found positive in the developed assay. Thus, the sensitivity, specificity, PPV, and NPV were found to be 100% (36/36), 97.7% (170/174), 90% (36/40), and 100% (170/170), respectively.

Figure 2 demonstrates a proposed strategy for the implementation of the LAMP-turbidimetric assay in molecular analysis. The cost-effectiveness and time consumed were compared between the conventional protocol, a strategy with LAMP-colorimetric assay, and a novel strategy with LAMP-turbidimetric assay using 210 samples. As depicted in Fig. 2, the cost of using conventional gap-PCR without the LAMP assay would be USD 3,507 (210 × 16.7). In contrast, a proposed strategy with LAMP assay with colorimetric detection would cost USD 2,558 [LAMP (210 × 9) and gap-PCR (40 × 16.7)]. The novel proposed strategy with LAMP assay using the turbidimetric assay would be USD 899 [LAMP (210 × 1.1) and gap-PCR (40 × 16.7)]. The reduction in cases requiring DNA analysis with gap-PCR would be 170 out of 210 (80.96%) when using the proposed strategy with the LAMP assay.

Discussion

In Thailand, the prevention and control program for severe thalassemia was launched in 1993 (Yamsri et al., 2010). The three diseases covered by this program include homozygous α0-thalassemia, homozygous β0-thalassemia, and compound heterozygous β0-thalassemia/Hb E. In this study, we implemented the prevention and control program in accordance with the announcement from the Ministry of Public Health  (Yamsri et al., 2010; Fucharoen & Winichagoon, 2007). The results of the retrospective study indicated successful couples screening at 84.25%, with the remaining participants being single pregnant women. For single women who tested positive, husbands were invited to undergo screening as per the program’s protocol. Out of 307 single pregnant women, 143 (46.58%) were deemed not to require further investigation due to negative screening. However, couple screening is preferable to single screening as it can reduce the number of negative-screened cases by about 75.51% in this region. Positive-screened cases proceed to the next step of Hb analysis investigation. When comparing cost-effectiveness, single screening appears to be less expensive than couple screening. However, the risk assessment procedure must include the step of inviting husbands when a positive screening occurs during pregnancy. This limitation may result in delayed risk assessments (Fucharoen et al., 2004; Yamsri et al., 2010).

Hb analysis served as the second screening for investigating fetal risk. While Hb analysis can diagnose the types of β0-thalassemia carriers, β0-thalassemia disease, or Hb E, it generally cannot be used to investigate α0-thalassemia carriers, except for cases with the Hb type EA (when Hb E > 25%). This Hb type allows the exclusion of couples from the risk of homozygous α0-thalassemia, based on the cut-off guideline (Hb E > 25%) (Jomoui et al., 2017a; Sanchaisuriya et al., 2003). This cut-off is employed in the prevention and control program to reduce the workload for further investigation of α0-thalassemia through DNA analysis. In Table 2, homozygous α0-thalassemia (Hb Bart’s hydrops fetalis syndrome) represents the most common risk of severe thalassemia and is subsequently confirmed with DNA analysis. In a previous study in northeast Thailand, compound heterozygous β0-thalassemia/Hb E was found to be predominant (66%) in fetal risk (Yamsri et al., 2010). In northeast Thailand, Hb E has a high prevalence, reaching up to 50% in this region (Jomoui et al., 2017b). While three types of severe thalassemia are a concern in Thailand, the major problem may vary in different regions based on thalassemia gene frequency, as reported in previous studies. Thus, studying thalassemia gene prevalence should be conducted alongside the implementation of prevention and control programs (Yamsri et al., 2010; Fucharoen & Winichagoon, 2007; Nopparatana et al., 2020; Pansatiankul & Saisorn, 2003).

The conventional DNA analysis for the detection of α0-thalassemia was still performed despite the screening tests being conducted, resulting in a high workload. This study identified different phenotypes and genotypes that required investigation through DNA analysis. However, Table 3 shows a positive rate of 18.10% (38/210) and the remaining false positives from the previous screening at 81.90%. This outcome increased the workload for DNA analysis and incurred expensive costs in the investigation. Pre-DNA analysis screening has been reported in several strategies, such as IC strip assay and isothermal amplification. However, the IC strip assay still demonstrated high costs and false positives  (Jomoui et al., 2017a; Prayalaw, Fucharoen & Fucharoen, 2014; Winichagoon et al., 2015). On the other hand, isothermal amplification, especially the LAMP colorimetric assay that has been reported, exhibited high sensitivity and specificity. Nevertheless, cost-effectiveness remains a challenge as it is still considered expensive (Jomoui et al., 2022).

In this study, we successfully developed a low-cost LAMP-turbidimetric assay for detecting α0-thalassemia (SEA deletion). This technique is high-speed, does not require an expensive machine, and is cost-effective. We could develop the LAMP technique under constant temperature, and finally, the results can be observed with the naked eye through turbidity. This forms the basis of LAMP detection and is the most cost-effective when compared with other methods. In the study, the sensitivity and specificity were reported to be 100% and 97.7%, respectively. Previous studies using LAMP for detecting α0-thalassemia (SEA deletion) have reported colorimetric or pH-sensitive dye LAMP assays. Chomean et al. demonstrated their colorimetric LAMP assay for detecting α0-thalassemia (SEA deletion), where a positive reaction changed a colorless solution to a light blue color solution with 100% sensitivity and specificity (Chomean et al., 2018). Meanwhile, Jomoui et al. (2022) used isothermal temperature based on a colorimetric assay, changing a yellow solution to a pink color solution with 100% sensitivity and 98.2% specificity.

The main objective of this study was cost-effectiveness for routine use. We demonstrated cost-effectiveness in a scenario involving 210 samples that required DNA analysis, comparing a conventional strategy, a strategy with LAMP-colorimetric assay (Jomoui et al., 2022), and a novel strategy with LAMP-turbidimetric assay. We found that using LAMP-turbidimetric before DNA analysis was cheaper than the other two strategies. The total cost reduction for 210 samples was about USD 1,659 and USD 2,608 compared with LAMP-colorimetric assay and the conventional method, respectively. Furthermore, we could reduce the workload of DNA analysis with gap-PCR by 170/210 (80.96%) when using the proposed strategy with both LAMP assays. However, limitations were also observed. Four cases of false positives were noted in the developed assay, but these will be investigated with the gold standard gap-PCR in the final step, as per the proposed strategy. The false positive result may be attributed to the high sensitivity of the LAMP assay or cross-contamination between samples within the LAMP assay run. However, these four samples were repeated, and the results showed negativity in all cases in the second time. Additionally, the developed assay can only detect the SEA deletion of α0-thalassemia, while other rare α0-thalassemia variants may go undiagnosed. A limitation of the developed assay, it does not detect other α0-thalassemia such as THAI deletion, Filipino deletion, Chiang Rai deletion, 20.5 kb deletion, MED deletion. These mutations have a very low incident in this region. For improve the efficiency of prevention and control programs for severe thalassemia disease in the couple screening, we recommended that when found only one parent positive with the LAMP assays (SEA deletion) as the proposed in Fig. 2, this couple should be further investigation with the gold standard gap-PCR for all α0-thalassemia–would help in preventing a misdiagnosis of the couple with these rare α0-thalassaemia defects. In this study, validation was based on a single center. Recommendations for further studies include a larger population and considering different ethnicities or regions. Finally, the developed LAMP turbidity assay is more applicable before confirmation by the conventional PCR assay in situations of high demand and limitations in the workload of DNA analysis. Moreover, this assay could be conducted in community hospitals located in remote areas with limited advanced equipment, such as a thermo-cycler.

Conclusion

The prevention and control program for severe thalassemia in the central part of Thailand was demonstrated and proved successful as a policy. Additionally, a LAMP with a simple turbidity assay was developed. These developed LAMP assays are more cost-effective, convenient, simple, rapid, and do not involve complicated processes. Moreover, our proposed strategy to apply the LAMP turbidity assay would effectively support the thalassemia screening policy for detecting α0-thalassemia (SEA deletion).

Supplemental Information

Supplemental Information 1 Raw data with results LAMP turbidity

Supplemental Information 2 The calculation of cost per test

Additional Information and Declarations

Competing Interests

Author Contributions

Human Ethics

Data Availability

The authors declare there are no competing interests.

Wittaya Jomoui conceived and designed the experiments, performed the experiments, analyzed the data, prepared figures and/or tables, authored or reviewed drafts of the article, acquisition of the grants, and approved the final draft.

Kanokkorn Saknava conceived and designed the experiments, performed the experiments, analyzed the data, authored or reviewed drafts of the article, and approved the final draft.

Kanokpron Prechatrammaruch conceived and designed the experiments, performed the experiments, analyzed the data, authored or reviewed drafts of the article, and approved the final draft.

Yanticha Ondee conceived and designed the experiments, performed the experiments, analyzed the data, authored or reviewed drafts of the article, and approved the final draft.

The following information was supplied relating to ethical approvals (i.e., approving body and any reference numbers):

Ethical approval for this study was obtained from the Institutional Review Board of Srinakharinwirot University, Thailand (SWUEC/E/M-098/2565E).

The following information was supplied regarding data availability:

The raw data and the calculation of cost per test is available in the Supplementary Files.

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
