# Peer review of "Retrospective study and implementation of a low-cost LAMP-turbidimetric assay for screening α0-thalassemia (SEA deletion): preventing and controlling Hb Bart’s hydrops fetalis syndrome in Thailand"

_PeerJ, doi:10.7717/peerj.17054_

## Round 0.1 · original submission · Major Revisions

Dear Authors,

Please address the issues raised by the reviewers and resubmit your edited manuscript together with your answers to reviewers.

Reviewer 1 ·

Basic reporting

No comment.

Experimental design

No comment.

Validity of the findings

No comment.

Additional comments

The authors developed a low-cost LAMP-turbidimetric assay for screening α0-thalassemia (SEA deletion) in the prevention and control program of Hb Bart’s hydrops fetalis in Thailand. There are some questions as follows;
1. Are these in-house reagents for the developed LAMP-turbidimetric assay? What is 10X isothermal amplification buffer?
2. Please show the details to calculate the cost per test in the 3 methods (conventional PCR, LAMP colorimetric, and LAMP-turbidimetric assays)
3. What is the new knowledge in this study? This manuscript is very similar to “Jomoui W, Srivorakun H, Chansai S, Fucharoen S. Loop-mediated isothermal amplification (LAMP) colorimetric phenol red assay for rapid identification of α0-thalassemia: Application to population screening and prenatal diagnosis. PLoS One. 2022 Apr 28;17(4):e0267832.” I think the new finding is that the developed in-house LAMP-turbidimetric assay is cheaper than commercial kits of LAMP colorimetric assay.
4. The authors used both α-thalassemia 1 and α0-thalassemia. α-thalassemia 1 should be changed to α0-thalassemia in the whole manuscript.
5. Results, lines 166-168 and Table 1: how to define the subjects with normal MCV (MCV > 80fL) and reduced MCH (MCH < 27 pg), or reduced MCV (MCV < 80fL) and normal MCH (MCH > 27 pg)? Are these subjects positive screening results?

Reviewer 2 ·

Basic reporting

no comments

Experimental design

no comments

Validity of the findings

The findings of the manuscript are already validated.

Additional comments

Overall, the manuscript presents valuable information regarding the feasibility of using the LAMP turbidity assay to detect α0-thalassemia (SEA deletion) and proposes its implementation for the prevention and control of severe α0-thalassemia (SEA deletion) in Thailand. However, I think the authors should consider the following issues:
1. The introduction at the first of the manuscript needs more detail to provide more justification for this study. The authors need to clarify the sentence in lines 95-96 “This technique is particularly intriguing and offers several advantages compared to other reported LAMP assays”. What are these advantages?
2. The authors need to explain and demonstrate how they came up with the sample size of 210 samples for DNA analysis in lines 115-118.
3. A large novel 44.6 kb deletion named α0-thalassemia Chiang Rai (--CR) was reported in this study (Table 3). I suggest that the authors need to elaborate on the description of the thalassemia diagnosis in lines 121-128.
4. I strongly suggest the authors explain and clarify the false positive cases (n=4) in the discussion section.
5. In this study, the low-cost LAMP-turbidimetric assay to detect α0-thalassemia (SEA deletion) was proposed for routine settings and would improve the efficiency of prevention and control programs for severe thalassemia disease in this region. I strongly suggest the authors explain the other α0-thalassemia management, especially THAI deletion in Figure 2 and the discussion section.

Reviewer 3 ·

Basic reporting

This study demonstrated that a low-cost LAMP based test can be used for a0 -thalassemia (SEA deletion) in large sample pool (couples and pregnant women).

Experimental design

The article explains the research question well and discusses the need in the field.

Validity of the findings

Tables and figures are ok but there are many abbreviations in the article especially in the tables. Explanation of these abbreviations can be included in table legend or as footnotes.

Abbreviations for MCV, MCH, and DCIP should also be included in tables.

in line 150 and 205, PPV and NPV abbreviations were used, the full form of these terms can be added when they first appear in text.

---

## Round 0.2 · Major Revisions

The reviewers suggested major revisions to your manuscript. Please address all issues raised by the reviewers. In addition, reviewers raised concerns that there is a high overlap with your previous publication in PLOS ONE. Make sure to cite your previous paper and give a good reason why you need to publish this work as a separate article (i.e. explain how the papers differ and why you think there is a need to do the additional work).

Reviewer 1 ·

Basic reporting

No comment.

Experimental design

No comment.

Validity of the findings

No comment.

Additional comments

I have additional minor points as follows. If the authors respond to these points, I don't have any questions.
1. In Supplementary file 2, What are control materials? Positive and negative control? Why don't LAMP assay with colorimetric and LAMP assay with turbidimetric assay have control materials?
2. Magnesium pyrophosphate and 10X isothermal amplification buffer should be included in LAMP assay with turbidimetric assay in Supplementary file 2.

Reviewer 2 ·

Basic reporting

No comment

Experimental design

No comment

Validity of the findings

No comment

Additional comments

The authors have adequately improved their manuscript and thorough response to the comments from the previous review. However, there are a few important points that have to be clarified further as follows:
1. In response to Review 2, Comment 1, the authors need to provide the reference in lines 98-100 “Furthermore, colorimetric pH indicators change may be influenced by different DNA components sample, while turbidity is not affected.”
2. In response to Review 2, Comment 2, the authors need to add the reference regarding the sample size formula. They need to clarify and explain the proportion/ prevalence (P = 0.05) that was used to calculate in lines 119-121.
3. In response to Review 2, Comment 3, the authors need to clarify the sentence in lines 131-133 “For one case of rare α0-thalassemia (SEA deletion) Chiang Rai (--CR) was also investigated with convention multiplex gap-PCR described elsewhere.” Does the case have α0-thalassemia mutation (SEA deletion) or α0-thalassemia mutation (Chiang Rai deletion)?

Reviewer 3 ·

Basic reporting

This study demonstrated that a low-cost LAMP based test can be used for a0 -thalassemia (SEA deletion) in large sample pool (couples and pregnant women).

Experimental design

The article explains the research question well and discusses the need in the field.

Validity of the findings

The authors made the necessary changes in the manuscript and the manuscript is acceptable for publication.

---

## Round 0.3 · accepted · Accept

Dear Authors,

Reviewers found your manuscript acceptable for publication in its revised form.
Congratulations.

Please follow the journal's communications regarding the timely publication of your manuscript.

Sincerely.

Reviewer 2 ·

Basic reporting

No comment

Experimental design

No comment

Validity of the findings

No comment

Additional comments

The authors have sufficiently improved their manuscript.

Reviewer 3 ·

Basic reporting

No comment

Experimental design

No comment

Validity of the findings

No comment

Additional comments

The authors provided necessary explanation and the article is acceptable.